# Global Transcriptional and Epigenetic Reconfiguration during Chemical Reprogramming of Human Retinal Pigment Epithelial Cells into Photoreceptor-like Cells

**DOI:** 10.3390/cells11193146

**Published:** 2022-10-06

**Authors:** Xiaoqian Deng, Ryan Lee, Sin Yee Lim, Zheng Zhong, Jing Wang, Yizhi Liu, Guoping Fan

**Affiliations:** 1State Key Laboratory of Ophthalmology, Zhongshan Ophthalmic Center, Sun Yat-sen University, Guangdong Provincial Key Laboratory of Ophthalmology and Visual Science, Guangzhou 510060, China; 2Department of Human Genetics, David Geffen School of Medicine, University of California, Los Angeles, Los Angeles, CA 90095, USA; 3Department of NanoEngineering, University of California, La Jolla, San Diego, CA 92093, USA; 4Shanghai Institute for Advanced Immunochemical Studies, ShanghaiTech University, Shanghai 201210, China

**Keywords:** human retinal pigment epithelial cells, photoreceptor-like cells, chemical reprogramming, single-cell RNA sequencing, DNA methylation sequencing

## Abstract

Retinal degenerative diseases are frequently caused by the loss of retinal neural cells such as photoreceptors. Cell replacement is regarded as one of the most promising therapies. Multiple types of stem and somatic cells have been tested for photoreceptor conversion. However, current induction efficiencies are still low and the molecular mechanisms underlying reprogramming remain to be clarified. In this work, by combining treatment with small molecules, we directly reprogrammed human fetal retinal pigment epithelial (RPE) cells into chemically induced photoreceptor-like cells (CiPCs) in vitro. Bulk and single-cell RNA sequencing, as well as methylation sequencing, were performed to understand the transcriptional and epigenetic changes during CiPCs conversion. A multi-omics analysis showed that the direct reprogramming process partly resembled events of early retina development. We also found that the efficiency of CiPCs conversion from RPE is much better than that from human dermal fibroblasts (HDF). The small molecules effectively induced RPE cells into CiPCs via suppression of the epithelial-to-mesenchymal transition (EMT). Among the signaling pathways involved in CiPCs conversion, glutamate receptor activation is prominent. In summary, RPE cells can be efficiently reprogrammed into photoreceptor-like cells through defined pharmacological modulations, providing a useful cell source for photoreceptor generation in cell replacement therapy for retinal degenerative diseases.

## 1. Introduction

Photoreceptors are responsible for converting light into electrical signals for further processing and integration, which is essential for the visual system. Retinal degenerative diseases, such as age-related macular degeneration (AMD) and retinitis pigmentosa, are typically characterized by the degeneration of retinal pigment epithelial (RPE) cells and photoreceptors, causing irreversible vision loss [1,2,3]. Cell therapy has been extensively investigated to repair or replace damaged RPE cells or photoreceptors [4,5,6]. In particular, cell replacement therapy based on directed differentiation of embryonic stem cells (ESCs) [7] and induced pluripotent stem cells (iPSCs) [8] into photoreceptor cells holds tremendous promise for the treatment of retinal diseases [9]. However, the potential risk of generating progenitor cell tumors presents a challenge for their use in clinical therapy [10]. 

Recent studies also demonstrate the approach to generating photoreceptors from Müller glia and RPE cells [11,12] via viral-delivered transcription factors. Another promising method is to use small molecules to spatiotemporally regulate cell fate without disrupting genome integrity [13]. According to Mahato et al. [14], a five small molecule factor (5F) cocktail containing valproic acid (VPA, HDAC inhibitor), CHIR99021 (GSK3 inhibitor), RepSox (TGFR-inhibitor), Forskolin (cAMP activator), and IWR1 (Wnt/β-catenin pathway inhibitor) could chemically induce fibroblasts into photoreceptor-like cells (CiPCs). The CiPCs functionally restored pupil reflex and visual function after transplantation into the subretinal space of mice with retinal degeneration. Notably, different types of fibroblast cells exhibited variable but low conversion efficiencies (from 0 to 1%). Meanwhile, the molecular changes during 5F-induced cell fate conversion are still unclear. It is important to improve the low conversion efficiency and elucidate the underlying mechanisms when CiPCs are to be used in future clinical applications. 

The RPE and neural retina share the same optic vesicle origin during development [15] and are physically adjacent to each other. In amphibians, the RPE has an outstanding capacity to regenerate the retina and restore lost sight [15]. Though this phenomenon is absent in adult mammals, human RPE cells still possess the potential to be activated into stem-like cells and differentiate into multiple lineages [16,17]. Fetal RPE cells have native RPE properties and can be expanded in vitro into a large quantity [18], potentially providing a useful cell source for cell reprogramming and replacement therapy.

Here we report that fetal RPE cells are prone to being rapidly transformed into CiPCs upon sequential administration of small molecules and downregulation of the RNA-binding protein PTBP1, which was reported to promote neuronal conversion. The RPE-derived CiPCs exhibit favorable survival ability and express photoreceptor markers after subretinal injection. Transcriptome and methylome profiles of the CiPCs are similar to the native photoreceptors, single-cell RNA sequencing (scRNA-seq) analysis reconstructs the reprogramming path from RPE cells to CiPCs, suggesting important roles for epithelial-to-mesenchymal transition (EMT) suppression and glutamate receptor activation during reprogramming. In addition, we found that RPE cells exhibit a much higher efficiency for photoreceptor conversion than human dermal fibroblasts (HDF), indicating that RPE is a more desirable cell source for photoreceptor production in vitro.

## 2. Materials and Methods

### 2.1. Cell Culture

Primary fetal RPE cells (12–18 weeks of gestation) and HDF were collected as described in earlier work [19,20,21]. RPE cells were incubated in 0.1% gelatin-coated culture plates at 37 °C with 5% CO_2_. RPE medium was changed every other day, containing α-MEM (Gibco, Gaithersburg, MD, USA), L-glutamine (Sigma-Aldrich, St. Louis, MO, USA), 1X N1 supplement (Sigma-Aldrich, St. Louis, MO, USA), non-essential amino acids (Sigma-Aldrich, St. Louis, MO, USA), 0.25 mg/mL taurine (Sigma-Aldrich, St. Louis, MO, USA), 0.013 μg/L triiodo-thyronine (Sigma-Aldrich, St. Louis, MO, USA), 20 μg/L hydrocortisone (Sigma-Aldrich, St. Louis, MO, USA), 1X penicillin-streptomycin (Gibco, Gaithersburg, MD, USA), and 5% fetal bovine serum (FBS; Thermo Fisher Scientific, Waltham, MA, USA). The HDF cells were maintained in a growth medium containing DMEM (Gibco, Gaithersburg, MD, USA), 10% FBS, 1% L-glutamine, and 1% penicillin-streptomycin. Mycoplasma was not detected by using a PCR-based kit (Universal Mycoplasma Detection Kit, ATCC, Manassas, VA, USA).

### 2.2. Generation of CiPCs

Chemically, induction of photoreceptor-like cells through small molecules was conducted according to the methods previously described [14] with modifications. The primary HDF and RPE cells were used in the first five passages. RPE cells were seeded at a density of 3 × 10^4^ cells/cm^2^ and cultured in RPE medium on Day 0 (D0). The next day (D1), the medium was changed to photoreceptor induction medium (PIM) containing DMEM/F12 (Gibco, Gaithersburg, MD, USA), 10% knockout serum replacement (Gibco, Gaithersburg, MD, USA), 2% B27 (Gibco, Gaithersburg, MD, USA), 5 ng/mL Noggin (PeproTech, NJ, USA), and 5 ng/mL insulin-like growth factor 1 (IGF1; PeproTech, cranbury, NJ, USA), in combination with five small molecule factors (5F): 0.5 mM VPA (Sigma-Aldrich, St. Louis, MO, USA), 4.8 μM CHIR99021 (Cayman Chemical, MI, USA), 2 μM Repsox (Cayman Chemical, MI, USA), 10 μM Forskolin (Cayman Chemical, MI, USA), and 10 μM IWR1 (Sigma-Aldrich, St. Louis, MO, USA). To promote and support the formation of photoreceptors, 3 nM sonic hedgehog (R&D Systems, MN, USA), 100 μM taurine, and 1 μM retinoic acid (Sigma-Aldrich, St. Louis, MO, USA) (STR) were added along with 5F from D5 to D10. Cells were also maintained in photoreceptor differentiation medium (PDM) containing DMEM/F12, 2% B27, 1% N2 (Gibco, Gaithersburg, MD, USA), 10 ng/mL Noggin, 10 ng/mL IGF1, and 5 ng/mL basic fibroblast growth factor (bFGF; PeproTech, NJ, USA) with 5F to D22. For HDF reprogramming, cells were seeded in HDF growth medium. On D1, PIM containing 5F (at the same concentrations) was added and the subsequent process was the same as that of RPE cells mentioned above. The medium was changed every two days. The mediums and reagents used are listed in Appendix A.

### 2.3. Treatment with ASO

To test if PTBP1 ASO could improve conversion efficiency, we used the mouse PTBP1 ASO synthesis and transfection protocols previously published [22]. PTBP1 ASO was obtained by Integrated DNA Technologies (Northridge, CA, USA). The sequence of the target region in human PTBP1 for ASO synthesis is 5′-GGGTGAAGATCCTGTTCAATA-3′. The backbone of the ASO contains phosphorothioate modification. Fluorescein was attached to the 3′ end of the ASO.

Cells were seeded on D0 and cultured in the growth medium to reach confluency at around 70–80% 24 h later. On D1, in addition to a medium change of PIM with 5F, 75 pmol of PTBP1 ASO was transfected with Lipofectamine RNAiMAX (Thermo Fisher Scientific, Waltham, MA, USA). Forty-eight hours later, the medium was replaced by fresh PIM containing 5F, then the following process was the same as that of treatment with 5F alone after D3.

### 2.4. Time-Lapse Imaging

For live-cell time-lapse imaging, synchronized RPE cells were plated on glass-bottom dishes coated with 0.1% gelatin. The next day, the culture medium was replaced with PIM plus 5F, then image recording was conducted on the first day of 5F treatment in a temperature- and CO_2_-controlled chamber. At least 6 positions per well were acquired every 30 min with a Zeiss Axio Observer Z1 inverted fluorescence microscope.

### 2.5. RT–qPCR

Total RNA was extracted using the PureLink™ RNA Mini Kit (Thermo Fisher Scientific, Waltham, MA, USA), and 1 μg RNA was reverse transcribed to cDNA with a High-Capacity cDNA Reverse Transcription Kit (Applied Biosystems, Waltham, MA, USA). The total RNA concentration was determined by the NanoDrop Spectrophotometer (Thermo Fisher Scientific, Waltham, MA, USA). Real-time qPCR was carried out with PowerUp™ SYBR™ Green Master Mix (Applied Biosystems, Waltham, MA, USA) on a CFX96 real-time PCR detection system (Bio-Rad, Hercules, CA, USA). Fold changes were calculated using the 2^−ΔΔCt^ method. In order to improve the data accuracy, we adopted *GAPDH* and *ACTB* as reference genes and normalized the data by taking the geometric mean. Primers are listed in Appendix A.

### 2.6. Immunofluorescence

Cells were cultured on coverslips for photoreceptor induction. Experiments with different combinations of conditions (control, ASO only, 5F, and 5FA) were performed at the same time as one replicate. Slides were washed with phosphate buffered saline (PBS; Sigma-Aldrich, St. Louis, MO, USA) before fixation in 4% paraformaldehyde at room temperature for 20 min. After three washes, the cells were blocked in 5% bovine serum albumin (BSA; Sigma-Aldrich, St. Louis, MO, USA) with 0.3% Triton-X 100 (Sigma-Aldrich, St. Louis, MO, USA) at room temperature for 1 h. Primary antibodies were incubated overnight at 4 °C. After three washes with PBS, the slides were incubated with secondary antibodies for 1 h at room temperature, then washed three times and mounted with DAPI Fluoromount-G (Southern Biotech, Birmingham, AL, USA). Images were taken on an Eclipse 80i Nikon microscope and analyzed using ImageJ (v2.1.0). The antibodies used are provided in Appendix A.

### 2.7. Trypan Blue Exclusion Assay

Cell viability was estimated using the trypan blue exclusion assay. Viable cells have intact membranes to exclude trypan blue, while dead cells lose membrane integrity and can thus be stained by this dye. Cells with 5F/5FA induction at different time points were dissociated by incubation with 0.25% trypsin-EDTA (Sigma-Aldrich, St. Louis, MO, USA) at 37 °C for 5 min, centrifuged and then resuspended in the culture medium. A trypan blue solution was added to the cell suspensions in a ratio of 1:1. Total cells and dead cells (stained in blue) were counted using a hemocytometer. The percentage of living cells was calculated.

### 2.8. Western Blots

RPE cells, reprogramming intermediates, and CiPCs were harvested and dissociated in a lysis buffer. After protein extraction, samples were loaded onto 12% gels and separated via SDS-PAGE (sodium dodecyl sulfate polyacrylamide gel electrophoresis). Then the proteins were transferred to polyvinylidene difluoride (PVDF) membranes (Bio-Rad, Hercules, CA, USA), blocked in 5% skim milk (Sigma-Aldrich, St. Louis, MO, USA) at room temperature for 1 h, and incubated with primary antibodies at 4 °C overnight. The membrane was transferred into secondary antibodies the next day and incubated for 1 h at room temperature. The specific signals were detected with ECL Western Blotting Substrate (Thermo Fisher Scientific, Waltham, MA, USA) and visualized with a chemiluminescence detection system (Bio-Rad, Hercules, CA, USA). Signal intensity was analyzed with ImageJ (v2.1.0). The antibodies used are provided in Appendix A.

### 2.9. Establishment of Mouse Model and Cell Transplantation

All animal work was conducted in accordance with the guidelines of the University of California, Los Angeles Animal Research Committee (ARC). 8–10-week-old C57B6L/J mice were maintained under a 12 h light/12 h dark illumination cycle with normal food and water. Anesthesia was induced via intraperitoneal injection of ketamine (87.5 mg/kg) and xylazine (12.5 mg/kg) before each surgical procedure. To establish retinal degeneration, we employed the widely used sodium iodate (NaIO_3_) mouse model [23]. Unlike the genetic retinal degeneration models such as rd1 mouse with RPE unaffected [24], NaIO_3_ leads to RPE and subsequent photoreceptor degeneration, which better recapitulates the advanced state in dry AMD patients. The procedures for NaIO_3_ model establishment and cell transplantation were adjusted from published articles [17,25,26]. Briefly, 35 mg/kg of sterile 1% NaIO_3_ (Sigma-Aldrich, St. Louis, MO, USA) in saline was administered through a tail vein injection. One month after NaIO_3_ treatment, 5F/5FA treated RPE cells at D10 were individualized by incubation at 37 °C for 5 min with 0.25% trypsin-EDTA and resuspended into single-cell suspension with balanced salt solution (BSS; Alcon, Fort Worth, TX, USA). The cells were counted on a hemocytometer to reach a dilution of 1 × 10^5^ cells/μL and kept on ice until subretinal transplantation. Briefly, 1% tropicamide ophthalmic solution was used to dilate the pupil. A tiny bleb was visible below the retina on the superior nasal location by injecting 1.5 μL of 1 × 10^5^/μL cell suspension or BSS through the sclera approximately 1 mm behind the limbus under a surgical microscope using a 33-gauge blunt-end microliter syringe (Hamilton, Reno, NV, USA) as previously described [27]. The contralateral eye received the same treatment. The animals were monitored for revival from anesthesia and ocular inflammation.

Selected mouse eye frozen sections were washed three times in PBS before being blocked in 5% BSA with 0.5% Triton-X 100 for two hours at room temperature. Then, the process was followed by incubation with primary antibodies at 4 °C overnight. After three subsequent PBS washes, secondary antibodies were incubated for 1 h at room temperature. Sections were mounted with DAPI Fluoromount-G (Southern Biotech, Birmingham, AL, USA). Images were taken on an Eclipse 80i Nikon microscope and analyzed using ImageJ (v2.1.0). Primary and secondary antibodies used in this article are provided in Appendix A.

### 2.10. RNA Sequencing (RNA-seq) and Data Processing

Total RNA was extracted using a PureLink™ RNA Mini Kit (Thermo Fisher Scientific, Waltham, MA, USA). Then mRNAs were used for library construction by using the NEBNext^®^ Ultra™ II RNA Library Prep Kit (New England Biolabs, Ipswich, MA, USA) following the manufacturer’s recommendations. Libraries were quantified using the 2200 TapeStation (Agilent, Santa Clara, CA, USA) and sequenced using the Illumina HiSeq 3000 or Novaseq S2.

All the sequenced raw reads were filtered and trimmed with Trim Galore (v0.4.1) before subsequent analysis. For bulk RNA-seq data processing, the clean reads were mapped to the human genome (GRC38) using Hisat2 [28]. Then, the number of reads uniquely mapped to each gene was determined using FeatureCounts from subread2 [29]. The differentially expressed genes (DEGs) were identified by DESeq2 [30] with p values less than 0.01. Principal component analysis (PCA) was performed using all the genes after removing the batch effect by DESeq2.

### 2.11. Gene Ontology (GO) and Gene Set Enrichment Analysis (GSEA)

GO analysis was carried out by the GENEONTOLOGY platform [31]. GSEA was performed on all the datasets in MSigDB (v7.4) [32,33].

### 2.12. Enzymatic Methylation Sequencing (EM-seq) and Data Processing

Genomic DNA of different cells was extracted using DNeasy Blood and Tissue Kits (Qiagen, Valencia, CA, USA). Genomic DNA was spiked with 0.5% lambda DNA and fragmented to 250–350 bp using Bioruptor (Diagenode, Denville, NJ, USA). EM-seq libraries were prepared from 300 ng of fragmented DNA using the NEBNext Enzymatic Methyl-seq Kit (New England Biolabs, MA, USA). 

For EM-seq data processing, reads were filtered and trimmed with Trim Galore (v0.4.1) followed by mapping to the human genome (GRC38) using Bismark (v0.17.0) [34]. Mapped reads were further deduplicated and filtered for non-conversion. Estimation of methylation levels was determined in the CpG context with Bismark. 

### 2.13. 10× scRNA-Seq and Data Processing

For scRNA-seq, the preparation of single cells was performed according to protocols from 10× Genomics. Cells were trypsinized into single cell suspensions and resuspended in an appropriate buffer with viability of over 90%. Then cell suspensions were introduced into 10× Chromium for single-cell 3′ transcriptome profiling. 

The resulting FASTQ files were processed using CellRanger v3, using GRC38 as the reference genome. The read count matrix generated by CellRanger was then analyzed using Seurat v3 [35]. Cells were further filtered by the number of genes detected (with at least 500 genes but no more than 6000 detected) and the percentage of reads mapped to the mitochondrial genome out of total reads (less than 10%). The top 2000 genes were identified by variable feature selection based on a variance stabilizing transformation (“VST”). Then 50 principal components (PCs) were utilized to calculate the k-nearest neighbors (KNN) graph based on the Euclidean distance in PCA space. Clusters were then visualized using a Uniform Manifold Approximation and Projection (UMAP) plot to annotate the cell types by gene markers. Genes with a log2 (fold change of expression) of at least 0.25 and FDR < 0.01 were selected as DEGs. Cell cycle phase assignments were performed using the Seurat package. Monocle2 was used for pseudo-time analysis [36].

To comprehensively evaluate the conversion rate of a related cell fate, the AddModuleScore function of the Seurat v3 R package was used to calculate the Rod score and Cone score, respectively. For Rod score calculation, the “RCVRN”, “PDE6G”, “VSX2”, “CRX”, “OTX2”, “RHO”, “NRL”, “PAX6”, “ASCL1”, “RXRG”, “RORB”, and “THRB” were used. For Cone score calculation, “GNAT2”, “OPN1SW”, and “OPN1LW” were used. For each sample, this calculates the average expression of genes in the module, subtracted by the average expression of a randomly selected set of control genes with similar expressions across the samples. As with input to the function, we used the normalized expression as described above, and in each case, we used 100 random control genes.

### 2.14. Statistical Analysis

All data were presented in the form of mean ± standard deviation (SD). For comparisons between multiple groups, an analysis of variance (ANOVA) followed by a Tukey’s or Bonferroni post hoc test was used. Comparisons between two groups used the Student’s t-test. For comparison between groups with unequal variances, a non-parametric Mann–Whitney test was used to compare two groups. Replicates were obtained by measuring distinct donor samples (3 biological replicates), and statistical analyses were based on at least 3 experimental replicates. Details of the number of biological replicates and p values are provided in the figure legends. For all analyses, a significance level α = 0.05 was set with a 95% confidence level, and differences were considered significant at a *p* value ≤ 0.05.

## 3. Results

### 3.1. Direct Conversion of RPE Cells into CiPCs with Small Molecules

It is very instructive that fibroblasts could be directly and chemically reprogrammed into functional CiPCs by five small molecule factors (5F), but the efficiency was rather low [14]. We first tested the effect of cell source on the conversion process, and both HDF and RPE cells were used for 5F-mediated CiPCs induction (Figure 1a). For HDF, no obvious morphological changes were observed at the early stage (Day (D) 3) of induction (Figure 1b). After 10 days (D10) (Figure 1b), HDF were reprogrammed into CiPCs (H-CiPCs), which was consistent with the previous study. For RPE cells, both bright-field images and live-cell time-lapse imaging showed that they gained dramatic morphological changes as early as the first day of reprogramming (Figure 1b,c, Appendix A) and exhibited an intermediate state (D3). On D10, most of the RPE-derived CiPCs (R-CiPCs) appeared as single cells with round soma, among which were scattered cells with tubular structures and small colonies (Figure 1b). Extended cultures showed that cells were able to be maintained in photoreceptor differentiation medium (PDM) to D22 (Appendix A). These results reflect a faster response of RPE cells to 5F compared to HDF.

Photoreceptor-related genes such as *RHO*, *RCVRN,* and *CRX* were significantly increased in both H-CiPCs and R-CiPCs at D10 (Figure 1d). We further compared the reprogramming efficiency between HDF and RPE cells as donor cells by immunostaining. At D3, about 35.2% ± 6.5% (*p* < 0.001) of RPE-derived reprogramming intermediates (RI) co-expressed TUJ1 and MAP2, which are detected in the neuroretina lineage. Approximately, 33.5% ± 2.3% (*p* < 0.001) of cells co-expressed TUJ1 and CRX in RPE-derived RI (Figure 1e,f), whereas only a few HDF-derived RI expressed these markers (Appendix A). RPE cells expressed photoreceptor markers RHODOPSIN (RHO) and RECOVERIN (RCVRN) on D10 (Figure 1g,h), as confirmed by Western blot (Appendix A). RHO and RCVRN positive cells in R-CiPCs after 5F treatment were approximately 28.9% ± 4.3% and 17.7% ± 5.7%, respectively, and much higher than those in H-CiPCs, which were 9.1% ± 1.0% (*p* < 0.01) and 12.4% ± 1.5% (*p* < 0.05), respectively (Appendix A). Thus, RPE cells exhibit a better reprogramming efficiency during the CiPCs induction than HDF.

To explore the cell conversion rate of CiPCs beyond D10, we extended the cell culture to D15, when the CiPCs would mature. The conversion rates of H-CiPCs and R-CiPCs were roughly stable at D15 compared to D10, but the total cell number decreased dramatically (Appendix A). Thus, extended cell culture will not further improve the conversion rate in both H-CiPCs and R-CiPCs.

We also tested the small molecules individually and found that they failed to generate a good number of RCVRN-positive cells (Appendix A). Additionally, different seeding densities were also tested to determine the proper starting density. As shown in Appendix A, 3 × 10^4^ cells/cm^2^ was the most suitable seeding density for RPE cell reprogramming.

### 3.2. Suppression of PTBP1 Partly Enhances CiPCs Induction

PTBP1 is an RNA-binding protein and is reported to inhibit neuronal differentiation by suppressing the splicing of a subset of neural targets [37]. Recently, studies have shown remarkable efficiency in neuronal conversion by downregulating PTBP1, although this observation is still being debated [22,38,39,40]. Antisense oligonucleotides (ASO), a promising therapeutic agent, could bind sequences specifically to the target RNA and modulate protein expression [41]. To test whether repression of PTBP1 could improve the reprogramming efficiency, we combined PTBP1 ASO treatment (added from D1 to D3) with 5F (5FA) for CiPCs induction. We found that the morphological changes due to 5FA treatment were similar to those of 5F-induced H-CiPCs and R-CiPCs (Figure 2a,b). Compared to 5F, 5FA leads to downregulation of PTBP1 (Figure 2c,d) and elevated expression of rod photoreceptor-specific genes such as *RHO* and *RCVRN* at D10 (Figure 2e,f). Correspondingly, immunostaining analysis indicated that RHO and RCVRN positive cells in 5FA condition were significantly increased in both H-CiPCs (10.1% ± 1.2%, *p* < 0.01 and 13.7% ± 4.5%, *p* < 0.01) and R-CiPCs (38.3% ± 8.2%, *p* < 0.001 and 42.1% ± 6.0%, *p* < 0.001) (Figure 2g,h and Appendix A). 5FA-induced R-CiPCs also expressed CRX, but very few were positive for the cone photoreceptor marker OPN1SW (Appendix A).

Meanwhile, we also tested if suppression of PTBP1 alone is sufficient to induce reprogramming. As shown in Figure 2a,b, no obvious morphological change could be observed. In addition, photoreceptor markers remained at a relatively low expression level (Figure 2e–h, Appendix A). Our results indicate that PTBP1 suppression itself is not enough to reprogram HDF and RPE cells into photoreceptor-like cells in vitro. Collectively, these results reflect that PTBP1 ASO further promoted, in part, the conversion rate based on 5F treatment.

### 3.3. CiPCs Sustained the Photoreceptor-Like Features In Vivo

To explore the practical implications of R-CiPCs from a clinical perspective, we transplanted the cells into the subretinal space in a NaIO_3_-induced RPE and photoreceptor degeneration mouse model, which has a similar pathological process to dry AMD and has been widely used to test various retinal cell transplantation treatments [17,26] (Appendix A). As is shown in Appendix A, after 4 weeks of NaIO_3_ injection, the thickness of the retina’s outer nuclear layer decreased significantly. We collected the R-CiPCs after 10 days of induction for subretinal transplantation. Upon immunofluorescence staining analysis at 4 weeks post-transplantation, we observed that the transplanted cells survived well and maintained the characteristics of CiPCs in the subretinal space of the injected eyes in close apposition with the outermost layer of the host retina (inner nuclear layer and occasional patches of the remaining outer nuclear layer). We confirmed that these cells were of human origin by staining with human nuclear antigen antibody (HNA), and around 23% of them were RCVRN positive (Figure 2i,j, Appendix A). Collectively, these results suggest RPE cells could be a promising cell source for photoreceptor cell generation for retinal cell replacement therapy.

### 3.4. Small Molecules Effectively Reshaped the Transcriptional Profile of RPE Cells

To comprehensively investigate the molecular changes during reprogramming, RNA-seq was performed on the cells at the initial (D0, HDF, and RPE cells), early (D3, with 5F and 5FA induction), and late stages of reprogramming (D10, with ASO, 5F, and 5FA induction). Principal component analysis (PCA) distinguished each type of the CiPCs well based on their cell origins, and both followed a reprogramming path similar to fetal retina development (Figure 3a, Appendix A), revealing the effective reshaping of the transcriptional profile. In particular, the photoreceptor genes showed higher expression levels in R-CiPCs than in H-CiPCs (Figure 3b).

Then we compared the transcriptional profile of R-CiPCs at different stages to explore the mechanism of R-CiPCs induction. Comparing the early intermediates at D3 to RPE control, 1068 DEGs were detected (Figure 3c, Appendix A). GO analysis showed that the upregulated genes were enriched in neuronal genesis, neurotransmitter uptake, and photoreceptor cell differentiation, while the downregulated genes were enriched in regulation of epithelial cell migration, epithelium development, and cell−cell junction assembly (Figure 3d).

Notably, the neuronal genes such as *SOX8* and *IGFN1* showed increased expression but the epithelium-related genes such as *EGF* and *FGF2* decreased (Appendix A). In addition, photoreceptor-specific transcriptional factors, such as *ASCL1, RXRG, THRB,* and *RORB,* were upregulated in RI (Appendix A). These results reflected an emerging lineage reprogramming from epithelial cells to neuronal cells in the intermediate state.

At D10, the transcriptome profile revealed consistent upregulation of the rod-specific genes in R-CiPCs compared to RPE cells (Appendix A), but only a slight increase in cone-related gene expression (Appendix A). As shown in Figure 3e,f, the upregulated genes were enriched in photoreceptor cell differentiation while the downregulated genes were enriched in epithelium development. Consistently, GSEA revealed that both RI and R-CiPCs expressed enriched genes involved in photoreceptor differentiation and function, while the control RPE cells showed enrichment in cell substrate junction, which confirmed continuous conversion from RPE cells to photoreceptors (Figure 3g,h, Appendix A).

In addition, we found that RPE cells were prone to exhibiting epithelial-mesenchymal transition (EMT) with increased passage number, as previously reported (Appendix A). The RPE cells undergoing EMT became less differentiated and lost their lineage-specific features. Correspondingly, we noticed that EMT-like morphological changes can be effectively eliminated through 5F treatment. Consistent with our observations, RNA-seq results demonstrated that the expression of EMT-related genes was significantly downregulated in R-CiPCs compared with RPE cells (Appendix A), which was confirmed by RT-qPCR and GSEA (Appendix A). Further analysis identified a decrease in TGF-β pathway genes in R-CiPCs (Appendix A). Because TGF-β has been reported to induce EMT [42], 5F could suppress EMT through the inhibition of the TGF-β pathway.

ASCL1, a powerful pro-neural transcription factor, has been reported to stimulate the regeneration of retinal neurons from Müller glia [43]. A previous study has reported that NF-κB induced *Ascl1* during the induction of CiPCs [14]. Consistently, we found that the upregulated genes in the CiPCs were enriched in the positive regulation of the NF-kB pathway, accompanied by upregulation of *ASCL1* (Appendix A).

Glutamate is an essential neurotransmitter released by photoreceptors for retinal synaptic circuitry [44]. Activation of the glutamate receptor pathway has been reported to induce *ASCL1* expression [45]. Our results demonstrated that the glutamate receptor pathway-related genes were gradually upregulated in R-CiPCs compared to RPE cells, accompanied by upregulation of *ASCL1* (Appendix A). This result was also confirmed by RT-qPCR of *GRM4* and *GLUD2* and GSEA (Appendix A). Thus, our data implied the potential involvement of the glutamate receptor pathway in the photoreceptor lineage transition, which needs to be further elucidated.

### 3.5. Global DNA Methylation Remodeling during Direct Reprogramming of RPE Cells to CiPCs

To investigate DNA methylation changes due to reprogramming, base-resolution methylomes were generated from control RPE cells as well as the reprogrammed R-CiPCs (Appendix A). Remarkably, both 5F and 5FA induced genome-wide demethylation (Figure 4a). VPA was reported to induce DNA demethylation through the action of ten-eleven translocation (TET) family enzymes in an active DNA demethylation pathway in HeLa cells [46]. Thus, the global demethylation may result from the increased expression of TET family genes due to VPA treatment (Figure 4b). Of note, 4195 genes showed dramatic demethylation in their promoters (Figure 4c). GO analysis indicated these genes were functionally enriched in the regulation of glutamate secretion and neurogenesis (Figure 4d,e,h). Notably, we found that the majority (437 out of 632) of the DEGs with demethylated promoters exhibited a significant increase in gene expression (Figure 4g), especially for the genes related to neurogenesis (Figure 4i).

In addition to the global demethylation, we noticed 166 genes showed significantly increased DNA methylation in their promoters (Figure 4c). These genes were functionally enriched in epithelium development (Figure 4f,j), and their hypermethylation was accompanied by attenuated gene expression (Appendix A). The hypermethylation of epithelium lineage genes could be attributed to the increased expression of DNA methyltransferase 3 (DNMT3) family genes (Figure 4k). These results demonstrated the removal of epithelium lineage features, and the permission of photoreceptor features due to DNA methylation remodeling during reprogramming.

### 3.6. Single-Cell Analysis of Cell Populations during CiPCs Reprogramming

Since direct reprogramming is an unsynchronized process, traditional methods such as bulk RNA-seq of inhomogeneous cell populations cannot fully dissect the detailed changes in the reprogramming pathway. Therefore, scRNA-seq provides a greater opportunity to capture the transcriptional state and delineate the trajectory of direct reprogramming at a single-cell resolution. We conducted 10× genomics scRNA-seq at different time points during RPE cell reprogramming to CiPCs. Sequencing data were obtained from 23,661 individual cells at early and late reprogramming phases (D3 and D10) (Figure 5a), with 4378 median genes and 82,255 mean confidently mapped reads per cell. We projected all cells on a UMAP plot and identified five transcriptionally distinct clusters: RPE cells, retinal progenitor cells (RPCs), reprogramming intermediate 1 (RI1), reprogramming intermediate 2 (RI2), and CiPCs, which reveal a dynamic transcriptomic transition from the parental RPE cells (Figure 5b). We specified the cell types by their unique expression of specific known marker genes (Figure 5c). *COL4A1* and *MKI67* are well known to be highly expressed in RPE cells and RPCs, respectively, whereas *ANXA1*, *HMGA1,* and *NEAT1* are widely expressed in neural cell types [47,48,49,50]. In addition to RPE cells, a fraction of RPCs were also present in the control group, which is consistent with the previous study [51]. The major components of the 5F treated cells at D3 were RPE cells, RPCs, and RI1, while 5FA treated cells at D3 were mainly composed of RPE cells, RPCs, RI1, and RI2. Additionally, CiPCs made up most of the 5FA group at D10. Compared with the control group, the proportion of RPE cells exhibited a gradual decline in the 5F D3, 5FA D3, and 5FA D10 groups, reflecting the progressively increasing conversion rate (Figure 5d). Detailed analysis showed that around 46% of the CiPCs at D10 were rod-like cells, while only 1% committed to the cone photoreceptor fate (Figure 5e), which is consistent with the immunostaining result (Appendix A). Similar to the bulk RNA-seq result, glutamate receptor pathway genes were significantly upregulated in RI and CiPCs (Appendix A). To understand the molecular features of RI1, RI2, and CiPCs, we compared the transcriptome of these cell subtypes to RPE cells. The quantities of DEGs identified in RI1, RI2, and CiPCs were 211, 1098, and 859, respectively. Although these differences were highly significant, the cell conversion process was ongoing and modest. GO analysis showed that the DEGs in these subtypes were all enriched in neurogenesis, axon genesis, and neuron differentiation (Figure 5g–i). Thus, it is confirmed that the pharmacologic reprogramming induced RPE cells into a neuronal lineage, which can go further and be more mature.

Then we determined the cell cycle state for each subtype via the Cell-Cycle Scoring analysis based on the scRNA-seq data. A decreased proportion of cells in a particular cell cycle phase suggests a more rapid transition in that phase [52]. Multiple studies have established that the G1 phase length is short in self-renewing cells, and can prolong with cell differentiation [53,54]. As shown in Figure 5f, there were no RPCs in the G1 phase, revealing its highly proliferative feature. Compared to RPE cells, the RI1 and RI2 groups exhibited shorter G1 phases, which indicated their reentry into the cell cycle and proliferation during the induction process. With further induction, CiPCs showed extended G1 phases similar to RPE cells, implying they turned into a relatively stable condition. 

### 3.7. Trajectory and Pseudo-Time Analysis Identify Dynamic Pathways during CiPCs Reprogramming

To further delineate the cell transition pathways, we used the Monocle method to determine a pseudo-temporal order between cell types [36]. Monocle uses an algorithm to learn the sequence of gene expression changes that each cell must go through as part of a dynamic biological process. The pseudo-time trajectory is constructed by placing each cell at its proper position based on the gene expression changes. The pseudo-time trajectory axis indicated that the RI state was between RPE cells and CiPCs (Figure 6a–d). As the pseudo-time values increased, cells transited through the RI state and eventually progressed towards CiPCs (Appendix A).

Then, we identified 5626 genes differentially expressed along the pseudo-time (FDR < 0.01), including the typical photoreceptor marker genes (Figure 6e). GO analysis on the top 1000 pseudo-time-related DEGs revealed that these genes were enriched in glutamate homeostasis and nervous system development, which was also consistent with the bulk RNA-seq result (Figure 6f). Among them, RPE lineage-related genes such as *TYR*, *SERPINF1*, and *TTR* were downregulated over the pseudo-time progression (Figure 6g, Appendix A), and the rod photoreceptor-lineage related genes such as *RCVRN*, *CRX*, *NR2E3*, *PDE6G*, and *NRL* were gradually upregulated over pseudo-time (Figure 6h, Appendix A).

### 3.8. RPE-Derived CiPCs Resemble Photoreceptors in Human Fetal Retina

To investigate the similarity of CiPCs derived from fetal RPE cells and human developing photoreceptor cells, we further compared our scRNA-seq datasets with the results from human fetal retina (fRetina) at fetal day (FD) 125, when all major retinal cell types, including photoreceptors, are present [55]. Through integrating the datasets and projecting them onto a single UMAP plot, nine distinct clusters were obtained (Figure 7a,b). Among them, seven clusters were previously well-defined in fRetina, including retinal progenitor cells (RPCs), photoreceptors (PR), amacrine cells (AC), retinal ganglion cells (RGC), horizontal cells (HC), ON bipolar cells (ON BC), and OFF bipolar cells (OFF BC). The intermediates (RI1 and RI2) and RPE cells could not be distinguished well, while CiPCs could be subsequently divided into two groups: CiPCs-precursors (CiPCs-pre) and CiPCs-photoreceptors (CiPCs-PR). Compared with RPE cells and RI, CiPCs-PR were closer to the photoreceptor subtypes in fRetina with higher expression levels of photoreceptor marker genes (Figure 7c,d). Furthermore, this combination enabled us to identify the proportion of photoreceptor-like cells in the D10 samples (Figure 7e). Cell proportion analysis showed that photoreceptors (CiPCs-PR) constituted about 28.3% of all the cells (724 out of 2552) in the D10 sample.

### 3.9. Unique DNA Methylation Profiles in RPE Cells May Contribute to the Better Efficiency in Photoreceptor Induction Than That in Fibroblasts

To investigate the role of DNA methylation in determining the efficiency of reprogramming, methylome comparison was also conducted on H-CiPCs and R-CiPCs. Similar to R-CiPCs, H-CiPCs exhibited a low methylation level in the promoter of photoreceptor genes, reflecting the activation of these genes (Figure 8a–c). Genome-wide analysis of all of these genes showed that the DNA methylation pattern in fRetina was closer to R-CiPCs in most regions (Figure 8d). Among all the 27,535 gene promoters analyzed, 18,986 from R-CiPCs exhibited a closer methylation pattern to that of fRetina (Figure 8e). We further filtered the top 1000 promoters according to the methylation difference between H-CiPCs and R-CiPCs. Notably, the genes corresponding to these promoters were enriched in retinal development (Figure 8f). Further investigation of the methylation profiles of HDF and RPE cells showed that most of these DNA methylation differences already existed before 5F treatment (Figure 8b,c,f,g). As a result, the retained epigenetic memory of their cellular origin could contribute to the varying reprogramming efficiency for H-CiPCs and R-CiPCs.

## 4. Discussion

Chemical approaches lead to major advancements in cell reprogramming and regenerative medicine. For example, small molecule treatment has been reported to reprogram human somatic cells to pluripotent stem cells, reflecting the powerful potential for small molecules in cell reprogramming [56]. Here, we demonstrate that small molecules could efficiently convert RPE cells into postmitotic photoreceptor-like cells, providing another example of the usefulness of chemical induction in cell lineage conversion.

DNA methylation has been frequently attributed to an essential role in lineage transition [57]. Epigenetic regulators, such as the TET gene family and DNMT3 gene family, have been reported to jointly regulate promoter epigenetic landscapes. Consistently, this study identifies increased expression of all the genes in the TET and DNMT3 families, revealing their critical role in reshaping the DNA methylation landscape of CiPCs. Additionally, ASCL1, a known transcription factor, has been reported to induce promoter methylation of fibroblast-specific genes during reprogramming [58]. In general, the DNA methylation reconfiguration during chemical reprogramming is the collective outcome of epigenetic factors and transcription regulators.

The epigenetic memory of the original cells has strong effects on reprogramming efficiency. By stabilizing the original state of genes of the previous donor cell type, it impedes the conversion process and causes incomplete reprogramming to a different cell type [59]. Both RPE and photoreceptor cells are developed from the optic vesicle in the embryonic stage, suggesting that they may share similar epigenetic states. Our results demonstrated that RPE cells not only responded faster to the 5F treatment but also exhibited higher efficiency compared to HDF. These results suggest that similar epigenetic states between the original cells and induced cells could facilitate cell lineage transition during CiPCs reprogramming.

The RNA-binding protein PTBP1 has been considered to be a promising target for direct reprogramming of non-neuronal cells into functional neurons [38]. Recent in vivo studies showed that downregulation of PTBP1 could reprogram Müller glia into retinal ganglion cells with high efficiency [39]. However, these claims have been challenged for technical reasons and lack solid proof to demonstrate lineage switching [60]. Our results show that PTBP1 ASO coupled with 5F could improve the CiPCs induction efficiency, but repression of PTBP1 alone is not sufficient to reprogram RPE cells or HDF into photoreceptors, suggesting the PTBP1 pathway only plays an auxiliary role in CiPCs induction.

Recent research has also revealed that ESCs and iPSCs may be good sources for generating CiPCs. However, potential contamination of undifferentiated pluripotent stem cells or incomplete differentiation may result in tumorigenicity [61]. RPE cells are differentiated cells with no risk of developing tumors. Furthermore, exogenous RPE cells can be easily transplanted into the subretinal space between the RPE and outer segments of photoreceptors. This raises the possibility of future in situ reprogramming of subretinal transplanted RPE cells into photoreceptors for better integration into the retina structure in vivo.

RPE cells are involved in the transportation of nutrients, recycling of proteins, elimination of photoreceptor debris, and also secretion of some growth factors [62]. All of these functions enable RPE cells to play a critical role in regulating the microenvironment in the eye. In fact, photoreceptor loss in patients is often accompanied by dysfunction or death of RPE cells [63]. Therefore, it is also possible to transplant CiPCs and RPE cells simultaneously when both types of cells are produced in vitro. In this way, we not only treat the loss of photoreceptors but also reconstruct the subretinal microenvironment with both RPE cells and CiPCs.

Reprogramming efficiency is majorly determined by the effect of the small molecules as well as the intrinsic properties of the source cells. RPE cells in amphibians can regenerate the entire retina under an injury state, and several studies have shown that mammalian RPE cells can be reprogrammed into retinal progenitor cells and retinal neurons in vitro [64]. Its innate plasticity further assures the effectiveness of the treatment with small molecules. Since in vivo reprogramming holds tremendous promise in regenerative medicine, it would be of great interest to determine whether endogenous RPE cells can be converted into photoreceptor-like cells within the subretinal space in future studies.

## 5. Conclusions

Our study establishes an improved approach to generating CiPCs from direct reprogramming of somatic cells. RPE cells, as the source cell, exhibited superior conversion efficiency compared to HDF via five small molecule factors (5F). When the chemical treatment was applied, further downregulation of PTBP1 could promote conversion efficiency. Through multi-omics sequencing, we found that 5F treatment could induce global transcriptional and epigenetic remodeling for the epithelium-to-retinal neuron transition. More detailed analysis revealed that 5F could suppress EMT through inhibition of the TGF-β pathway but enhance CiPCs conversion through activation of the glutamate receptor pathway. Additionally, the preferable performance of RPE cells for CiPCs induction may benefit from their retinal lineage epigenetic modifications. Collectively, our study has significant implications for chemical approaches in the future clinical use of retinal cell therapy.

## Figures and Tables

**Figure 1 cells-11-03146-f001:**
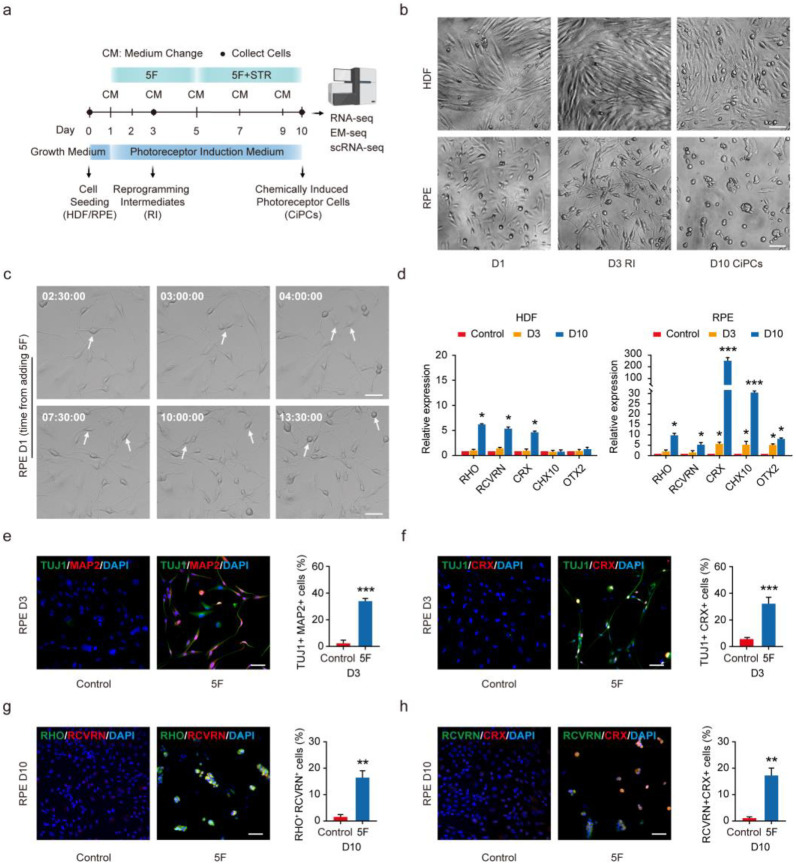
Conversion of HDF and human fetal RPE cells into CiPCs with defined small molecules. (**a**) Schematic of the chemical reprogramming process for CiPCs. 5F, five small molecule factors. STR, sonic hedgehog, taurine, and retinoic acid. (**b**) Bright-field images of HDF- and RPE-derived cells on Day (D) 1, D3 (RI), and D10 (CiPCs). Scale bar: 50 µm. (**c**) Representative images from live-cell time-lapse imaging show RPE cells undergoing morphological change within the first day after medium change with 5F. Time points were specified within the images. Arrow: typical cells with neural-like morphological change. Scale bar: 50 µm. (**d**) Relative mRNA expression of photoreceptor markers *RHODOPSIN* (*RHO*), *RECOVERIN* (*RCVRN*), *CRX,* and retinal progenitor markers *CHX10* and *OTX2* in HDF and RPE cells treated with small molecules at D3 and D10. *n* = four repetitions. Statistical significance was evaluated by a one-way ANOVA with a Tukey’s post hoc test. (**e**,**f**) Representative images of RPE cells treated with 5F co-stained for neuronal markers TUJ1, MAP2 (**e**) and TUJ1, CRX (**f**) at D3. Bar charts indicate the quantification of TUJ1 and MAP2 positive or TUJ1 and CRX positive cells among DAPI (4′, 6-Diamidino-2-phenylindole) positive cells at D3, respectively. Nuclei were stained with DAPI. Scale bars: 50 µm. (**g**,**h**) Immunofluorescence staining of RHODOPSIN (RHO), RECOVERIN (RCVRN) (**g**) and RCVRN, CRX (**h**) on RPE cells at D10 after 5F treatment. Quantifications of RHO and RCVRN or RCVRN and CRX positive cells are shown alongside. Nuclei were stained with DAPI. Scale bars: 50 µm. *n* = four for each group. e-h: Statistical comparison was performed by a two-tailed, unpaired t-test. * *p* < 0.05, ** *p* < 0.01, *** *p* < 0.001. Error bars indicate SD.

**Figure 2 cells-11-03146-f002:**
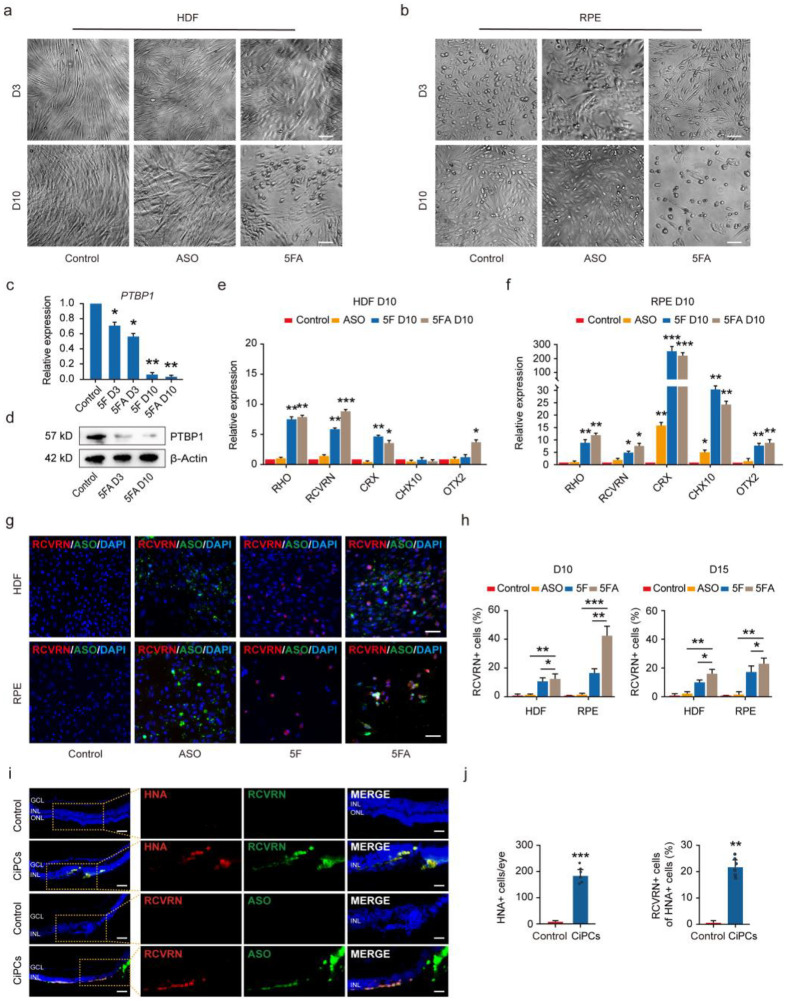
CiPCs induction is enhanced by suppression of PTBP1. (**a**,**b**) Bright-field images of HDF (**a**) and RPE cells (**b**) treated with PTBP1 ASO only or 5F plus ASO (5FA) at D3 and D10. Scale bars: 50 µm. (**c**) Relative mRNA expression of PTBP1 after 5F or 5FA treatment at D3 and D10. *n* = four repetitions. Statistical comparison was performed by a two-tailed, unpaired t-test. (**d**) Western blot analysis showing that the expression of PTBP1 was significantly downregulated after 5FA treatment at D3 and D10. *n* = three repetitions. (**e**,**f**) Relative mRNA expression of photoreceptor-related genes in ASO only, 5F, and 5FA treated HDF (**e**) and RPE cells (**f**) versus control at D10. *n* = four repetitions. Statistical significance was evaluated by one-way ANOVA with a Tukey’s post hoc. (**g**) Immunofluorescence staining of RCVRN on cells treated with ASO, 5F, or 5FA at D10. Nuclei were stained with DAPI. *n* = four repetitions of each group. Scale bars: 50 µm. (**h**) Quantification of RCVRN positive staining cells among all DAPI positive cells in different conditions at D10 and D15. Statistical significance was evaluated by two-way ANOVA with a Bonferroni post hoc. (**i**) Representative immunostaining images of CiPCs that survived into the subretinal space, stained for human nuclear antigen (HNA) and expressed the photoreceptor marker RCVRN. The dashed line boxes are magnified on the right side. GCL, ganglion cell layer; INL, inner nuclear layer; ONL, outer nuclear layer. Nuclei were stained with DAPI. Scale bars: left: 50 μm, right: 25 μm. (**j**) Quantification of average HNA positive cells per eye and RCVRN positive cells among all HNA positive cells. *n* = six eyes in each group. Statistical comparison was performed by a two-tailed Mann–Whitney test. * *p* < 0.05, ** *p* < 0.01, *** *p* < 0.001. Error bars indicate SD.

**Figure 3 cells-11-03146-f003:**
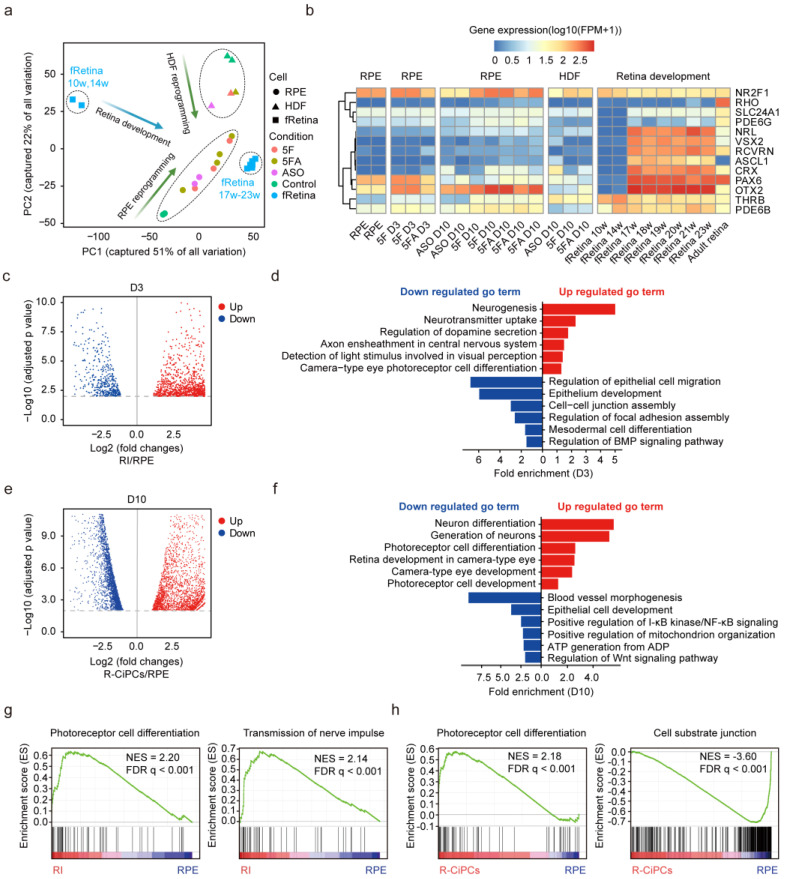
5F effectively reshaped the transcriptional profile of RPE cells. (**a**) Principal component analysis (PCA) based on gene expression of all the HDF-derived samples, RPE-derived samples, and fetal retina (fRetina) samples. W, week. (**b**) Heatmap showing the expression of photoreceptor marker genes in all samples, including fRetina, at different developing stages. (**c**,**d**) Identification of the differentially expressed genes (DEGs) in the reprogramming intermediates (RI) at D3 compared to RPE control (**c**) and corresponding gene ontology (GO) analysis (**d**). (**e**,**f**) Identification of the DEGs in the R-CiPCs at D10 compared to RPE control (**e**) and the corresponding GO analysis (**f**). (**g**) Gene-set-enrichment analysis (GSEA) plots evaluating the changes in the indicated gene signatures induced by the 5F treatment at D3. (**h**) GSEA plots, evaluating the changes in the indicated gene signatures induced by the 5F treatment at D10. NES, normalized enrichment score. FDR, false discovery rate.

**Figure 4 cells-11-03146-f004:**
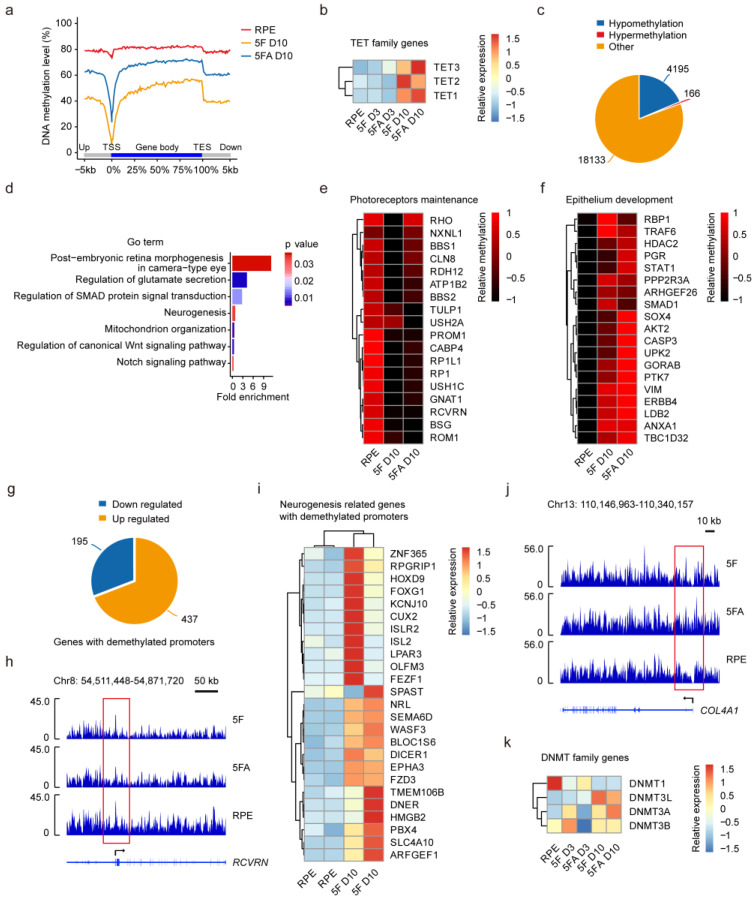
Global DNA methylation remodeling during direct reprogramming of RPE cells to R-CiPCs. (**a**) Line plot showing the average methylation level across the gene body. TSS, transcription start site. TES, transcription end site. (**b**) Heatmap showing gene expression changes of the TET family genes. (**c**) Pie plot showing the distribution of methylation changes at gene promoters upon 5F treatment. (**d**) GO analysis on the genes close to the dramatically demethylated regions due to CiPCs conversion. (**e**) Heatmap showing the level of methylation at the promoters of genes involved in photoreceptors maintenance. (**f**) Heatmap showing the level of methylation at the promoters of genes involved in epithelium development. (**g**) Pie plot showing the proportion of down- and upregulated DEGs that harbored hypomethylated promoters upon 5F treatment. (**h**) Example of genome browser view of DNA methylation profiles of *RCVRN* in R-CiPCs and RPE cells. (**i**) Heatmap showing the neurogenesis-related genes that were upregulated with hypomethylated promoters upon 5F treatment. (**j**) Example of genome browser view of DNA methylation profiles of *COL4A1* in R-CiPCs and RPE cells. Each track shows the percent methylation of individual CpGs. (**k**) Heatmap showing gene expression of the DNMT family genes during direct reprogramming.

**Figure 5 cells-11-03146-f005:**
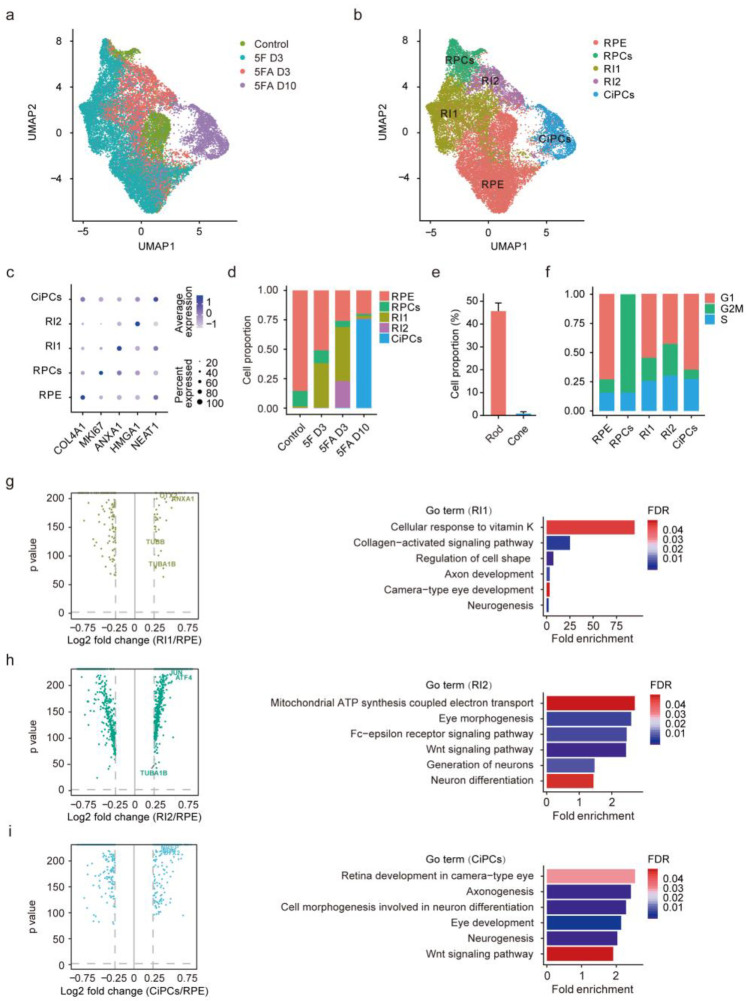
Single-cell analysis elicits homogeneous groups of cells during induction. (**a**,**b**) Uniform manifold approximation and projection (UMAP) analysis of single-cell transcriptomes from 23,661 cells annotated by treatment condition (**a**) and cell type (**b**), showing five cell clusters based on gene expression (each point represents a single cell). RPCs, retinal progenitor cells. (**c**) Dot plot showing marker genes identified in each cell subtype. (**d**) Bar plot showing the proportion of each cell subtype in different treatment conditions. (**e**) Bar plot showing the cell proportion of rod and cone photoreceptors inferred from scRNA-seq datasets. The proportion was determined by the percentage of cells with a positive rod score or cone score. (**f**) Bar plot showing the proportion of cell-cycle phases in each cell subtype. (**g**) Identification of the DEGs in RI1 compared with RPE (left) and the corresponding GO analysis (right). (**h**) Identification of the DEGs in RI2 compared with RPE (left) and the corresponding GO analysis (right). (**i**) Identification of the DEGs in CiPCs compared with RPE (left) and the corresponding GO analysis (right).

**Figure 6 cells-11-03146-f006:**
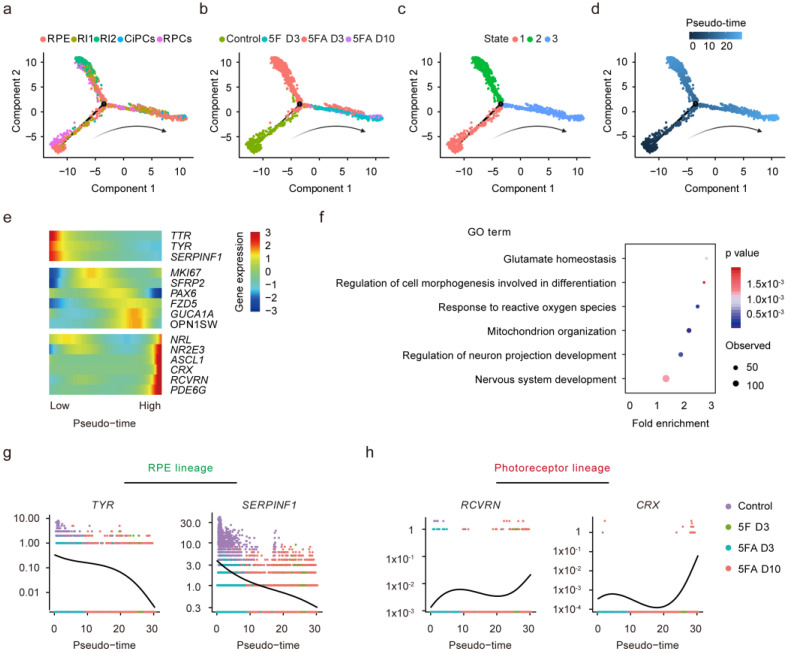
Trajectory and pseudo-time analysis identify dynamic changes along with the commitment to CiPCs. (**a**–**d**) Single-cell pseudo-time trajectory analysis using Monocle2. Cells on the tree are colored by cell type (**a**), treatment condition (**b**), state (**c**), and pseudo-time (**d**). The arrangement of cells on the trajectory from left to right (arrow) reveals the epithelium to retinal neuron transition. Cell states (**c**) are calculated by Monocle2 based on their pseudo-time values. (**e**) Heatmap of genes showing significant differential in pseudo-time-dependent expression. (**f**) GO analysis of genes showing significant differential pseudo-time-dependent expression. (**g**) Kinetic curves for RPE lineage genes *TYR* and *SERPINF1*. (**h**) Kinetic curves for photoreceptor lineage genes *RCVRN* and *CRX*.

**Figure 7 cells-11-03146-f007:**
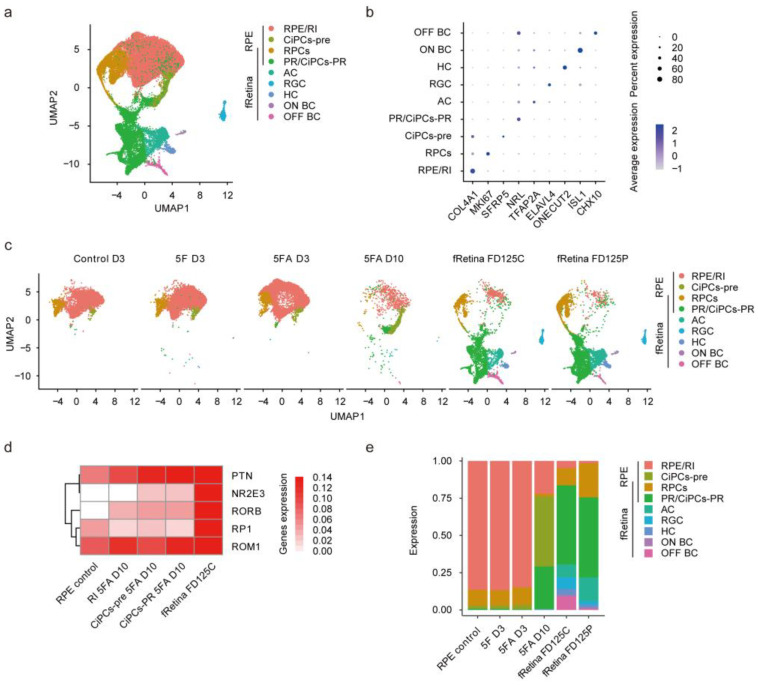
RPE-derived CiPCs resemble fetal retinal neurons. (**a**) UMAP analysis demonstrating clusters obtained by integrating analysis between single-cell RPE datasets and fRetina datasets. FD, fetal day. Cluster annotations: RI, RPE reprogramming intermediates; RPCs, retinal progenitor cells; CiPCs-pre, CiPCs-precursor, separate from fRetina; CiPCs-PR, CiPCs-photoreceptors, overlapping cells of CiPCs and fetal photoreceptors; PR, photoreceptors; AC, amacrine cells; RGC, retinal ganglion cells; HC, horizontal cells; ON BC, ON bipolar cells; OFF BC, OFF bipolar cells. (**b**) Dot plot showing the identified markers for each cell subtype. (**c**) UMAP analysis demonstrates the identified cell subtypes in different samples. fRetina D125C, human fetal retina at fetal day 125, central area. fRetina D125P, human fetal retina at fetal day 125, peripheral area. Integrating the scRNA-seq datasets of human fetal retina helped to define subgroups of CiPCs-pre and CiPCs-PR, thus the shape of the CiPCs cluster was different from that in Figure 5b (without fetal retina). (**d**) Heatmap showing the expression of photoreceptor marker genes in different samples. (**e**) Bar plot showing the proportion of each cell subtype identified in different samples.

**Figure 8 cells-11-03146-f008:**
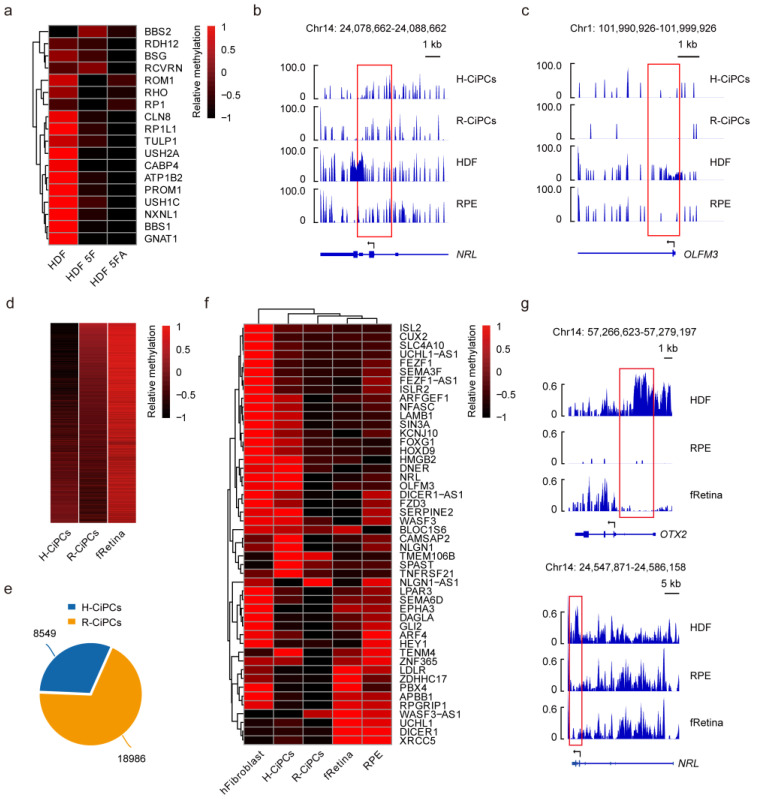
The retinal-like epigenetic modification enables RPE cells to perform better in photoreceptor induction. (**a**) Heatmap showing the differently methylated genes with the treatment of 5F or 5FA on HDF at D10. (**b**,**c**) Example of a genome browser view of DNA methylation profiles of *NRL* (**b**) and *OLFM3* (**c**) in HDF-derived samples and RPE-derived samples. Each track shows the percent methylation of individual CpGs. (**d**) Heatmap showing the differently methylated genes in H-CiPCs, R-CiPCs, and fRetina. (**e**) Pie plot showing the distribution of methylation changes in H-CiPCs and R-CiPCs at gene promoters upon 5F treatment. (**f**) Heatmap showing the differently methylated genes associated with neurogenesis in H-CiPCs and R-CiPCs. (**g**) Example of a genome browser view of DNA methylation profiles of *OTX2* and *NRL* gene in HDF, RPE, and fRetina. Each track shows the percent methylation of individual CpGs.

## Data Availability

All raw RNA-seq, EM-seq, and scRNA-seq data reported in this paper have been deposited in the National Center for Biotechnology Information Gene Expression Omnibus (GEO) database, at the accession number GSE192882. The RNA-seq and WGBS data of human fetal retina were obtained from GSE87064 [79]. The single-cell data from human fetal retina was obtained from GSE142526 [55]. The WGBS data of human fibroblasts were obtained from GSE8634 [80]. There are no restrictions on data availability, and all data will be made available upon request directed to the corresponding authors.

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
