# Peer review of "Global Transcriptional and Epigenetic Reconfiguration during Chemical Reprogramming of Human Retinal Pigment Epithelial Cells into Photoreceptor-like Cells"

_cells, 2022, doi:10.3390/cells11193146_

Round 1

Reviewer 1 Report

The paper by Fan and co-workers assesses the generation of chemically induced photoreceptor-like cells (CiPCs) from direct reprogramming of somatic cells. This is a large-scale study which is well designed and presented, and provides convincing evidence that retinal pigment epithelial cells could be converted to CiPCs and that this conversion is far more effective compared to human dermal fibroblasts. To induce remodeling authors employed a mixture of five small molecule factors  the effect of which can be enhanced by RNA-binding protein PTBP1. Together, those induce global transcriptional and epigenetic remodeling for the epithelium-to-retinal neuron transition.

The article can be printed in its present form, I would only suggest that the authors explain the appropriateness of the chemo-induced retinal degeneration method where this approach is mentioned for the first time in the article, namely in lines 180-182. It might also be worth describing in some detail why this model is preferable to the rd models with hereditary retinal degeneration.

Author Response

We thank the reviewer for the precious comments. We feel that these changes have further enhanced this manuscript.

Please see the attachment for the specific changes and responses.

Reviewer 2 Report

The manuscript entitled “Global transcriptional and epigenetic reconfiguration during chemical reprogramming of human retinal pigment epithelial cells into photoreceptor-like cells”by Deng and colleagues describe a massive dataset combining transcriptome and epigenetic analyses during the conversion of RPE into photoreceptor like cells. The studies are well-characterized and carefully executed, and the results are sure to be of profound interest to the research community. Before final acceptance, however, it is hoped that a few issues can be addressed: Otherwise, manuscript is well written with robust data.

Line 137: Only GAPDH is used for qPCR normalization. It would be more accurate If authors normalize the data using more than one reference gene by taking geometric mean. 

Line 430-431: Authors showed EMT associated genes in a heat map. It would be more informative If some of the key EMT genes (Example: SNAL, SLUG, TWIST1, ZEB1, CDH1, CDH2, VIM etc) are validated by qPCR or Western blot analysis. 

Author Response

We thank the reviewer for the precious comments. We feel that these suggestions and changes have further improved the manuscript.

Please see the attachment for the specific changes and responses.
